# Profiles Combining Muscle Atrophy and Neutrophil-to-Lymphocyte Ratio Are Associated with Prognosis of Patients with Stage IV Gastric Cancer

**DOI:** 10.3390/nu12061884

**Published:** 2020-06-24

**Authors:** Kota Shigeto, Takumi Kawaguchi, Shunji Koya, Keisuke Hirota, Toshimitsu Tanaka, Sachiko Nagasu, Masaru Fukahori, Tomoyuki Ushijima, Hiroo Matsuse, Keisuke Miwa, Koji Nagafuji, Takuji Torimura

**Affiliations:** 1Multidisciplinary Treatment Cancer Center, Kurume University Hospital, Kurume 830-0011, Japan; kotashigeto10072@gmail.com (K.S.); tanaka_toshimitsu@med.kurume-u.ac.jp (T.T.); shiraiwa_sachiko@med.kurume-u.ac.jp (S.N.); digdug0526@gmail.com (M.F.); ushijima_tomoyuki@kurume-u.ac.jp (T.U.); miwakeisuke@gmail.com (K.M.); knagafuji@med.kurume-u.ac.jp (K.N.); 2Division of Gastroenterology, Kurume University School of Medicine, Kurume 830-0011, Japan; tori@med.kurume-u.ac.jp; 3Division of Rehabilitation, Kurume University Hospital, Kurume 830-0011, Japan; kouya_shunji@kurume-u.ac.jp (S.K.); hirota_keisuke@kurume-u.ac.jp (K.H.); matsuse_hiroh@kurume-u.ac.jp (H.M.); 4Department of Orthopedics, School of Medicine, Kurume University, Kurume 830-0011, Japan; 5Division of Hematology and Oncology, Department of Medicine, Kurume University School of Medicine, Kurume 830-0011, Japan

**Keywords:** malnutrition, skeletal muscle, inflammation, stomach cancer, mortality

## Abstract

We aimed to investigate the impact of muscle atrophy and the neutrophil-to-lymphocyte ratio (NLR), a sub-clinical biomarker of inflammation and nutrition, on the prognosis of patients with unresectable advanced gastric cancer. We retrospectively enrolled 109 patients with stage IV gastric cancer (median age 69 years; female/male 22%/78%; median observational period 261 days). Independent factors and profiles for overall survival (OS) were determined by Cox regression analysis and decision-tree analysis, respectively. OS was calculated using the Kaplan–Meier method. The prevalence of muscle atrophy was 82.6% and the median NLR was 3.15. In Cox regression analysis, none of factors were identified as an independent factor for survival. The decision-tree analysis revealed that the most favorable prognostic profile was non-muscle atrophy (OS rate 36.8%). The most unfavorable prognostic profile was the combination of muscle atrophy and high NLR (OS rate 19.6%). The OS rate was significantly lower in patients with muscle atrophy and high NLR than in patients with non-muscle atrophy (1-year survival rate 28.5% vs. 54.7%; log-rank test *p* = 0.0014). In conclusion, “muscle atrophy and high NLR” was a prognostic profile for patients with stage IV gastric cancer. Thus, the assessment of muscle mass, subclinical inflammation, and malnutrition may be important for the management of patients with stage IV gastric cancer.

## 1. Introduction

Gastric cancer is a common malignancy worldwide and the third leading cause of cancer-related death [1]. Advances in diagnosis, surgical management, and chemoradiation therapy have contributed to an improvement in the prognosis of patients with early and locally advanced gastric cancer [2,3]. However, approximately 35% of patients with gastric cancer have distant metastases and have progressed to stage IV at the time of diagnosis [4]. In the REGATTA clinical trial, gastrectomy followed by chemotherapy did not demonstrate any survival benefit compared with chemotherapy alone in patients with stage IV gastric cancer [5]. Therefore, palliative chemotherapy remains the main therapeutic strategy for stage IV gastric cancer [6]. Despite recent developments in chemotherapy, the prognosis of patients with stage IV gastric cancer remains unsatisfactory [7].

Various factors are involved in the prognosis of advanced cancer. Visceral adiposity and fatty liver are risk factors for poor prognosis of patients with various cancers [8,9]. Glasgow Prognostic Score is a well-established parameter of inflammation [10]. In addition, the neutrophil-to-lymphocyte ratio (NLR) is a sub-clinical biomarker of inflammation and nutrition, in which the number of neutrophils is divided by the number of lymphocytes. NLR is not solely an index of inflammation, but is also known to reflect nutritional status, as the total lymphocyte count is decreased in cases of malnutrition [11,12]. A high NLR is associated with poor overall survival in patients with various tumors [13,14,15]. In addition, a higher NLR has been reported to be superior to the Glasgow Prognostic Score to predict the recurrence after curative surgery for patients with Stage II or III gastric cancer [16]. A high NLR is also associated with poor prognosis in patients with resectable gastric cancer [17]. Moreover, an increase in NLR during adjuvant chemotherapy indicates a poor prognosis in patients with stage II or III gastric cancer [18] and is associated with nutritional impairment and poor prognosis in patients with stage IV gastric cancer [19]. Thus, NLR is an important index for the management of patients with gastric cancer of any stage.

Sarcopenia is the loss of skeletal muscle mass and strength [20,21] and is seen in approximately 40% of patients with various types of advanced cancers [22]. Sarcopenia is associated with a poor prognosis in patients with esophageal cancer, colon cancer, and pancreatic cancer [23,24,25]. In patients with gastric cancer who underwent gastrectomy, sarcopenia is associated with severe post-operative complications [26,27]. Sarcopenia is also an independent risk factor associated with decreased survival in patients who undergo radical gastrectomy for gastric cancer [28]. Recently, sarcopenia has been reported as a prognostic factor for survival in patients with locally advanced gastroesophageal adenocarcinoma [29]. In addition, muscle atrophy assessed by CT is reported to predict overall survival in patients with gastric cancer who undergo gastrectomy [30]. However, the impact of muscle atrophy remains unclear in patients with stage IV gastric cancer.

The aim of this study was to investigate the impact of NLR and muscle atrophy on the prognosis of patients with stage IV gastric cancer.

## 2. Materials and Methods

### 2.1. Study Design

The purpose of this single-center retrospective observational study was to investigate the prognostic profiles for patients with stage IV gastric cancer.

### 2.2. Ethics

The study protocol conformed to the ethical guidelines of the Declaration of Helsinki, and prior approval was obtained from the institutional review board of Kurume University, Japan. An opt-out approach was used to obtain informed consent from the patients.

### 2.3. Patients

We enrolled 109 consecutive patients with stage IV gastric cancer between April 2014 and June 2019. The patients met the following inclusion criteria: (1) were 20 years of age or older; (2) had a performance status of grade 0–2; (3) were treated with systemic chemotherapy; and (4) underwent a biochemical examination and abdominal computed tomography (CT) scans. Exclusion criteria were as follows: (1) a performance status of grade 3–4; (2) severe heart, pulmonary, renal, or brain failure; (3) patient suffered from inflammatory, endocrine, or gastrointestinal diseases resulting in secondary sarcopenia; and (4) had other malignancies and malabsorption.

### 2.4. Diagnosis, Tumor Node Metastasis (TNM) Staging, and Treatment of Gastric Cancer

Gastric cancer was diagnosed by a pre-chemotherapeutic biopsy. The clinical stage of gastric cancer was evaluated by TNM staging based on the Japanese gastric cancer treatment guidelines of the Japanese Gastric Cancer Association [6]. Patients were treated with chemotherapy using S–1, oxaliplatin, cisplatin, 5-fluorouracil, capecitabine, and/or trastuzumab according to the Japanese gastric cancer treatment guidelines [6].

### 2.5. Prevalence of Comorbidities

Using medical records, we examined the prevalence of diabetes mellitus, hypertension, hypercholesterolemia, and hypertriglyceridemia before chemotherapy. The diagnosis of each co-morbidity was based on the published national guidelines [31,32,33].

### 2.6. Biochemical Examinations

Blood was obtained after overnight fasting and biochemical tests were performed. Serum levels of aspartate aminotransferase, alanine aminotransferase, lactate dehydrogenase, alkaline phosphatase, gamma-glutamyl transpeptidase, total protein, albumin, total bilirubin, total cholesterol, triglycerides, blood urea nitrogen, creatinine, estimated glomerular filtration rate, carcinoembryonic antigen, and CA19–9 were determined. We also measured fasting blood glucose, hemoglobin A1c, prothrombin activity, red blood cell count, hemoglobin level, white blood cell count, and platelet count.

### 2.7. Evaluation of NLR

Lymphocyte and neutrophil counts were obtained from the pre-chemotherapeutic blood examination and the NLR was calculated by dividing the neutrophil count by the lymphocyte count [11,12]. All subjects were classified into a high or low NLR group, as defined by the median value (median NLR, 3.15).

### 2.8. Evaluation of Skeletal Muscle Mass

Skeletal muscle mass was measured using CT imaging of the third lumbar vertebra, as previously described [34,35]. The CT images used for the measurement were performed for the assessment of gastric cancer before chemotherapy. Skeletal muscle mass was evaluated by the skeletal muscle index (SMI), which was calculated by normalizing the skeletal muscle area at the third lumbar vertebra by the square of the patient’s height (m^2^) [36,37]. Intramuscular adipose tissue content was calculated as the lumbar muscle-to-fat attenuation ratio, as previously described [38]. These analyses were performed using the diagnostic software ImageJ Version 1.50 (National Institutes of Health, Bethesda, MD, USA) [39].

In this study, sex-specific cutoff values of SMI were used to define muscle atrophy. Sakurai et al.’s previously published cutoff values were 43.2 cm^2^/m^2^ in males and 34.6 cm^2^/m^2^ in females, and these values were derived from the lowest sex-specific quartile of SMI in a Japanese gastric cancer cohort [30]. All subjects were classified into a non-muscle atrophy or muscle atrophy group according to the sex-specific cutoff values.

### 2.9. Evaluation of Visceral Fat Area and Hepatic Fat Content

Visceral fat area and hepatic fat content were assessed using CT scan images obtained before chemotherapy. Visceral fat area at the umbilical level is known to significantly correlate with total visceral abdominal fat volume [40,41]. Therefore, slices at the umbilical level were used for the evaluation of visceral fat area, as previously described [38]. Visceral adiposity was diagnosed as ≥100 cm^2^ of visceral fat area [42]. Hepatic fat content was calculated using the liver-to-spleen ratio: liver attenuation value/spleen attenuation value. The attenuation values were the average of the four measurements in each segment of the liver and spleen, as previously described [43]. Fatty liver was diagnosed as liver-to-spleen ratios <0.9 [44]. These analyses were performed using the diagnostic software ImageJ Version 1.50 (National Institutes of Health, Bethesda, MD, USA) [39].

### 2.10. Follow-up after Treatment with Chemotherapy for Gastric Cancer

After treatment with chemotherapy, patients were followed-up at regular intervals by routine physical examinations, biochemical tests, CT, and/or magnetic resonance imaging (MRI), according to the Japanese gastric cancer treatment guidelines, until death or the study censor date (12 June 2019) [6]. The median observation period was 261 days (range: 19–1348 days). All subjects were classified into the Alive or Deceased groups.

### 2.11. Safety Evaluation and Assessment of Adverse Event

Adverse events were assessed based on the National Cancer Institute Common Terminology Criteria for Adverse Events, version 4.0 [45]. In accordance with the guidelines, the dose of chemotherapy was reduced or treatment interrupted when any adverse event of grade 3 or higher severity, or any unacceptable drug-related adverse event, occurred.

### 2.12. Clinical Outcome

The primary endpoint of this study was the overall survival (OS) of the patients.

### 2.13. Statistical Analysis

Data are expressed as the median (interquartile range (IQR)), range, or frequency. The differences in continuous and categorical variables between the Alive and Deceased groups were analyzed by Wilcoxon rank sum test and chi-square test, respectively. In addition, independent factors and profiles associated with the mortality were analyzed using a Cox regression analysis with a step-wise variable selection and a decision-tree analysis, respectively [46,47]. Overall survival analysis was performed using the Kaplan–Meier method followed by a log-rank test. The level of statistical significance was set at *p* < 0.05. All statistical analyses were performed using JMP^®^14 software (SAS Institute Inc., Cary, NC, USA).

## 3. Results

### 3.1. Patients Characteristics

The median age of study subjects was 69 years and 78.0% of subjects were male (Table 1). All subjects had stage IV gastric cancer; 68.9% of patients were HER2 positive, and 56.9% of patients were treated with S–1/capecitabine/5-fluorouracil. Grade 3 adverse events were seen in 74.3% of patients, and neutropenia was a major adverse event, which was seen in 35.8% of patients.

Visceral adiposity and fatty liver were seen in 20% and 2% of patients. Biochemical parameters were summarized in Appendix A. The median NLR was 3.15, and the prevalence of muscle atrophy was 82.6% (Table 1). The median survival time for all subjects was 261 days, and the 1-year, 2-year, and 3-year survival rates were 39.1%, 21.6%, and 7.6%, respectively.

There was no significant difference in age, sex, body mass index (BMI), or performance status between those in the Alive and Deceased groups (Table 2). The prevalence of non-cardiac gastric cancer was significantly higher in the Alive group than in the Deceased group. No significant difference was seen in the chemotherapy regimen, prevalence of HER2 positive, or pathological diagnosis of gastric cancer between the Alive and Deceased groups. There was no significant difference in the prevalence of grade 3 adverse events and neutropenia between the Alive and Deceased groups.

There was no significant difference in the prevalence of visceral adiposity between the two groups. The prevalence of fatty liver was significantly higher in the Alive group compared to the Deceased group. Patients in the Deceased group had a significantly higher NLR than those in the Alive group. There was no significant difference in the prevalence of muscle atrophy and intramuscular adipose tissue content between the Alive and Deceased groups. In the Deceased group, serum albumin levels were significantly lower and blood hemoglobin A1c levels were significantly higher than those in the Alive group (Table 2). There was no significant difference between the two groups in other biochemical parameters (Appendix A).

### 3.2. Cox Regression Analysis with a Stepwise Variable Selection for Mortality

NLR and muscle atrophy were selected as variables for the Cox regression analysis by a stepwise variable selection. However, these factors were not statistically associated with mortality (Table 3). Other factors including age, sex, BMI, location of gastric cancer, location of gastric cancer, treatment regimen for gastric cancer, visceral adiposity, fatty liver, serum albumin level, and blood HbA1c level were not selected by a stepwise variable selection. Thus, there was no single independent factor associated with mortality in this study.

### 3.3. Decision-Tree Algorithm for Mortality

A decision-tree algorithm is a data-mining technique that reveals a series of classification rules by identifying priorities, and therefore, allow us to disclose a combination of important factors for the mortality of patients. To investigate profiles associated with mortality, a decision-tree analysis was performed. The decision-tree algorithm was created using two variables and patients were classified into three groups. Muscle atrophy was the first divergence variable, and the OS rate was 36.8% in patients in the non-muscle atrophy group (Profile 1, Figure 1), compared with an OS rate of 23.3% in patients with muscle atrophy. Among patients with muscle atrophy, NLR was the second divergence variable, and the OS rate was 19.6% in patients with muscle atrophy and high NLR (Profile 3 in Figure 1), compared to an OS of 27.2% in patients with muscle atrophy and low NLR (Profile 2, Figure 1). Other factors including age, sex, BMI, location of gastric cancer, treatment regimen for gastric cancer, visceral adiposity, and fatty liver were not identified as a divergence variable in a decision-tree analysis.

### 3.4. Differences in the OS among Each Profile Based on Decision-Tree Analysis for Mortality

We further investigated the impact of profiles according to the results of decision-tree algorithm for survival. Profiles 2 and 3 were identified as independent factors for mortality in the Cox regression analysis (Profile 2, OR 2.9321, 95%CI 1.0088–8.5220, *p* = 0.9481; Profile 3, OR 6.8024, 95%CI 1.8069–25.6093, *p* = 0.0046). Kaplan–Meier analysis was used to compare the OS rate among each profile based on the decision-tree analysis for mortality. In Profile 1, the OS rates were 54.7% and 27.3% at 1 and 2 years, respectively (Figure 2). In Profile 2, the OS rates were 43.2% and 30.5% at 1 and 2 years, respectively (Figure 2), while in Profile 3, the OS rates were 25.7% and 0.0% at 1 and 2 years, respectively (Figure 2). The OS rate of the Profile 3 was significantly lower than that of Profiles 1 and 2 (median 192 vs. 284 and 333 days, *p* = 0.0014) (Figure 2).

### 3.5. Cox Regression Analysis with a Stepwise Variable Selection for Muscle Atrophy

We performed a Cox regression analysis and high NLR and age were identified as independent factors associated with muscle atrophy (Table 4).

## 4. Discussion

In this study, we demonstrated that “muscle atrophy combined with high NLR” was a profile associated with OS in patients with stage IV gastric cancer. The survival time was significantly shorter in patients with muscle atrophy and high NLR compared to that of patients without muscle atrophy or with muscle atrophy and low NLR. Thus, assessment for muscle mass, subclinical inflammation, and malnutrition may be important for the management of patients with advanced gastric cancer.

In this study, the median participant age was 69 years, the median BMI was 20.3, and the median NLR was 3.15. These data are similar to those in previous studies for patients with stage IV gastric cancer [7,48,49]. In addition, the median survival time was 261 days in this study, comparable to the median survival time of less than 1 year that has been reported in patients with stage IV gastric cancer treated with chemotherapy [7,49,50]. Thus, the characteristics of patients enrolled in our study were similar to those in previous studies.

In our study, visceral adiposity and fatty liver were not identified as an independent prognostic factor. Neither NLR nor muscle atrophy were independent prognostic factors. Thus, there was no single independent variable associated with survival. However, Gonda et al. previously reported that elevated NLR is an independent prognostic factor in patients with stage IV gastric cancer [19]. Although it remains unclear why NLR was not identified as an independent prognostic factor in this study, a possible explanation is that in the previous study, 20.0% of enrolled patients showed no distant metastasis of gastric cancer [19]. However, only 2.8% of the patients enrolled in our study showed no distant metastasis. NLR has been reported to be associated with the prognosis of patients with earlier stages of gastric cancer (stage I, II, and III) rather than stage IV gastric cancer [19,51,52]. Thus, patients in our study may have a more advanced disease than those in the previous study and, therefore, various factors, rather than a single factor, are thought to be intricately associated with prognosis in this study.

No single factor was associated with mortality in this study. Therefore, a decision-tree analysis was employed to intricately assess profiles associated with survival of patients with stage IV gastric cancer. We initially identified the presence of muscle atrophy and high NLR as the primary and secondary divergence variables associated with survival, respectively. In addition, survival time was significantly shorter in patients with a profile combining muscle atrophy and high NLR than in patients with non-muscle atrophy or with muscle atrophy and low NLR. It remains unclear why muscle atrophy was identified as the most important variable for survival. It has been previously reported that muscle atrophy predicts toxicity from chemotherapy in stage IV gastric cancer patients [53,54]. In our study, all enrolled subjects were treated with chemotherapy. Muscle atrophy may be associated with chemotherapy-related toxicity, resulting in a poor prognosis.

In this study, a high NLR was ranked as the second prognostic variable after muscle atrophy in patients with stage IV gastric cancer. Although NLR has previously been reported as an independent prognostic factor in patients with stage IV gastric cancer [19,55,56], muscle atrophy was not evaluated in previous studies. Feliciano et al. reported that muscle atrophy combined with inflammation nearly doubled the risk of death in patients with colorectal cancer [57]. Muscle atrophy accompanied by a high NLR has also been associated with poor prognosis in patients with small cell lung cancer [58]. Thus, muscle atrophy and high NLR may synergistically worsen prognosis in patients with stage IV gastric cancer.

Any interaction between muscle atrophy and NLR remains unclear. However, high NLR, along with aging, was identified as an independent factor associated with muscle atrophy in this study. NLR is known as an indirect marker of poor nutrition status, which leads to sarcopenia [11,59]. In addition, Borges et al. recently reported that high NLR values were independently associated with sarcopenia in hospitalized patients with cancer [60]. Taken together, these data indicated a probable interaction between muscle mass and NLR.

This study has several limitations. This was a retrospective study conducted in a single center, and we enrolled patients with both initial and recurrent gastric cancer. In addition, there was heterogeneity among the enrolled subjects in the form of differing pathological diagnoses and regimens of chemotherapy. Finally, enteral nutrition was reported to improve short-term survival in patients with stage IV gastric cancer [61]; however, we did not evaluate the effects of nutritional therapy on prognosis. Thus, the prognostic importance of a profile combining muscle atrophy and high NLR should be evaluated in a multicenter prospective cohort study with a homogenous cohort of patients with stage IV gastric cancer.

## 5. Conclusions

In conclusion, we showed that a profile combining muscle atrophy and high NLR was associated with OS in patients with stage IV gastric cancer. The prognosis was significantly worse in patients with muscle atrophy and high NLR compared to that of patients with non-muscle atrophy or with muscle atrophy and low NLR. Thus, the assessment of muscle mass, subclinical inflammation, and malnutrition may be important for the management of patients with advanced gastric cancer.

## Figures and Tables

**Figure 1 nutrients-12-01884-f001:**
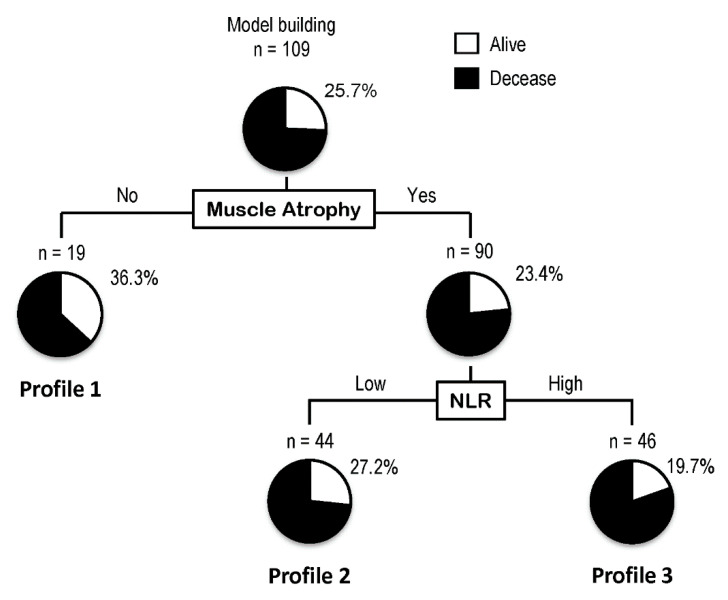
Profiles associated with survival in patients with stage IV gastric cancer. Decision tree algorithm for mortality. Pie graphs indicate the percentages of alive (white)/deceased (black) patients in each group. Abbreviations: NLR, neutrophil-to-lymphocyte ratio.

**Figure 2 nutrients-12-01884-f002:**
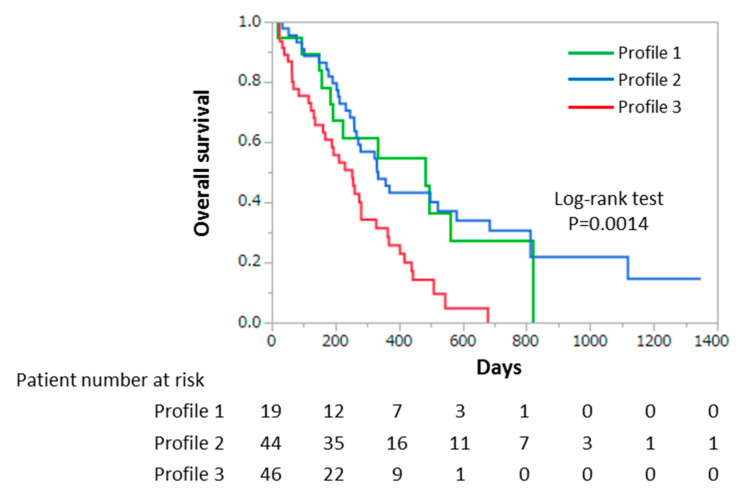
Overall survival time in patients with stage IV gastric cancer. Kaplan–Meier survival analysis shows the overall survival time according to Profiles 1, 2, and 3 in patients with stage IV gastric cancer. The green line represents Profile 1. The blue line represents Profile 2. The red line represents Profile 3.

**Table 1 nutrients-12-01884-t001:** Patient characteristics.

Variable	Reference Value	Median (IQR)	Range (Min–Max)
Number (*n*)	N/A	109	N/A
Age (Years)	N/A	69 (65–76)	55–85
Sex (Female/Male)	N/A	22.0%/78.0% (24/85)	N/A
Body mass index (kg/m^2^)	N/A	20.34 (18.7–22.6)	14.9–34.8
Performance status (0/1/2/3/4)	N/A	35.8%/54.1%/10.1%/0%/0% (39/59/11/0/0)	N/A
Gastric cancer (Initial/Recurrence)	N/A	80.7%/19.3% (88/21)	N/A
Location (Cardiac/Non cardiac)	N/A	29.4%/70.6% (32/77)	N/A
HER2 (Positive/Negative/Unknown)	N/A	65.1%/29.4%/5.5% (71/32/6)	N/A
Treatment of gastric cancer SOX/SP/FOLFOX S-1/Cape/5-FU SOXT/SPT/CAPOXT/XPT Others	N/A	19.2% (21/109) 56.9% (62/109) 15.6% (17/109) 8.3% (9/109)	N/A
Grade 3 adverse event (Yes/No)		74.3%/25.7% (81/28)	
Neutropenia (Yes/No)		35.8%/64.2% (39/70)	
Pathological diagnosis of gastric cancer (tub/por/sig/muc/others)	N/A	47.7%/34.9%/12.8%/1.8%/2.8% (52/38/14/2/3)	N/A
Presence of distant metastasis (Yes/No)	N/A N/A	97.2%/2.8% (106/3)	N/A N/A
Presence of diabetes mellitus (Yes/No)	N/A	35.8%/64.2% (39/70)	N/A
Presence of hypertension (Yes/No)	N/A	31.2%/68.8% (34/75)	N/A
Presence of hypercholesterolemia (Yes/No)	N/A	24.5%/75.5% (26/80)	N/A
Presence of hypertriglyceridemia (Yes/No)	N/A	42.0%/58.0% (58/80)	N/A
NLR	0.86–2.77	3.15 (2.30–5.27)	0.61–22.67
Presence of muscle atrophy (Yes/No)	N/A	82.6%/17.4%/ (90/19)	N/A
Intramuscular adipose tissue content	N/A	−0.52 (−0.61–−0.46)	−4.61–−0.013
Visceral fat area (cm^2^)	N/A	53.5 (26.3–92.2)	4.36–237.6
Visceral adiposity (Yes/No)	<100 cm^2^	20.0%/80.0% (20/82)	
Liver-to-spleen ratio	N/A	1.20 (1.092–1.34)	0.83–2.16
Fatty liver (Yes/No)	0.9	2%/98% (2/102)	

Note: Data are expressed as median (interquartile range (IQR)), range, or frequency. Abbreviations: IQR, interquartile range; N/A, not applicable; SOX, S-1 with oxaliplatin; SP, S-1 with cisplatin; FOLFOX, 5-fluorouracil with oxaliplatin; Cape, capecitabine; 5-FU, 5-fluorouracil; SOXT, S–1 with oxaliplatin and trastuzumab; SPT, S–1 with cisplatin and trastuzumab; CAPOXT, capecitabine with oxaliplatin and trastuzumab; XPT, capecitabine with cisplatin and trastuzumab; tub, tubular adenocarcinoma; por, poorly differentiated adenocarcinoma; sig, signet-ring cell carcinoma; muc, mucinous adenocarcinoma; NLR, neutrophil-to-lymphocyte ratio.

**Table 2 nutrients-12-01884-t002:** Comparison of patients’ characteristics between the Alive and Deceased groups.

Variable	Alive		Deceased		
	Median (IQR)	Range (Min–Max)	Median (IQR)	Range (Min–Max)	*p*
Number (N)	28	N/A	81	N/A	
Age (Years)	71 (65–76)	57–82	69 (65–76)	55–85	0.4832
Sex (Female/Male)	17.9%/82.1% (5/23)	N/A	23.5%/76.5% (19/62)	N/A	0.5376
Body mass index (kg/m^2^)	20.74 (18.7–22.4)	15.4–25.5	20.34 (18.3–22.7)	14.9–34.8	0.8487
Performance status (0/1/2/3/4)	46.4%/53.6%/0%/0%/0% (13/15/0/0/0)	N/A	32.1%/54.3%/13.6%/0%/0% (26/44/11/0/0)	N/A	0.0821
Gastric cancer (Initial/Recurrence)	66.7%/33.3% (18/9)	N/A	85.2%/14.8% (69/12)	N/A	0.0652
Location (Cardiac/Non cardiac)	14.3%/85.7% (4/24)	N/A	34.6%/65.4% (28/53)	N/A	0.0422
HER2 (Positive/Negative)	60.7%/39.3% (17/11)	N/A	72%/28% (54/21)	N/A	0.2708
Treatment of gastric cancer SOX/SP/FOLFOX S-1/Cape/5-FU SOXT/SPT/CAPOXT/XPT Others	67.8% (19) 3.6% (1) 14.3% (4) 14.3% (4)	N/A	36.7% (40) 18.3% (20) 11.9% (13) 8.3% (8)	N/A	0.0882
Grade 3 of adverse event (Yes/No)	78.6%/21.4% (22/6)		77.8%/27.2% (59/22)		0.5496
Neutropenia (Yes/No)	46.4%/53.6% (13/15)		32.1%/67.9% (26/55)		0.1727
Presence of diabetes mellitus (Yes/No)	21.4%/78.6% (6/22)	N/A	40.7%/59.3% (33/48)	N/A	0.661
Presence of hypertension (Yes/No)	39.3%/60.7% (11/17)	N/A	28.4/%/71.6% (23/58)	N/A	0.2836
Presence of hypercholesterolemia (Yes/No)	14.8%/85.2% (4/23)	N/A	27.9%/72.2% (22/57)	N/A	0.1742
Presence of hypertriglyceridemia (Yes/No)	0%/100% (0/27)	N/A	9.0%/91.0% (7/71)	N/A	0.1071
NRL	2.52 (1.87–4.38)	0.61–8.29.	3.17 (2.47–5.79)	1.01–22.67	0.0371
Presence of muscle atrophy (Yes/No)	75.0%/25.0% (21/7)		85.2%/14.8% (69/12)		0.2207
Intramuscular adipose tissue content	−0.54 (−0.62–−0.48)	−1.26–−0.32	−0.51 (−0.59–−0.46)	−4.61–−0.013	0.2539
Visceral fat area (cm^2^)	59.2 (43.7–101.1)	9.4–172.1	49.1 (22.2–92.2)	4.36–237.6	0.1674
Visceral adiposity (Yes/No)	25.9%/27.1% (7/20)		17.3%/82.7% (13/62)		0.3985
Liver-to-spleen ratio	1.20 (1.08–1.35)	0.83–1.46	1.12 (1.09–1.32)	0.95–2.16	0.9686
Fatty liver (Yes/No)	8.0%/92% (2/23)		0.0%/100% (0/76)		0.0128
LDH(IU/L)	183 (162–263)	123–499	226 (173–428)	106–2,323	0.0458
Albumin (g/dL)	3.81 (3.29–4.06)	2.00–4.40	3.47 (3.10–3.70)	1.80–4.85	0.0313
HbA1c (%)	5.7 (5.4–6.0)	4.9–7.4	5.9 (5.6–6.4)	3.9–10.9	0.0405

Note: Data are expressed as median (interquartile range (IQR)), range, or frequency. Abbreviations: IQR, interquartile range; SOX, S–1 with oxaliplatin; SP, S–1 with cisplatin; FOLFOX, 5-fluorouracil with oxaliplatin; Cape, capecitabine; 5-FU, 5-fluorouracil; SOXT, S–1 with oxaliplatin and trastuzumab; SPT, S–1 with cisplatin and trastuzumab; CAPOXT, capecitabine with oxaliplatin and trastuzumab; XPT, capecitabine with cisplatin and trastuzumab; tub, tubular adenocarcinoma; por, poorly differentiated adenocarcinoma; sig, signet-ring cell carcinoma; muc, mucinous adenocarcinoma; AST, aspartate aminotransferase; ALT, alanine aminotransferase; LDH, lactate dehydrogenase; ALP, alkaline phosphatase; GGT, gamma-glutamyl transpeptidase; BUN, blood urea nitrogen; eGFR, estimated glomerular filtration rate; HbA1c, hemoglobin A1c; CEA, carcinoembryonic antigen; NLR, neutrophil-to-lymphocyte ratio.

**Table 3 nutrients-12-01884-t003:** Logistic regression analysis with a stepwise variable selection for the factors associated with mortality.

Factors	Odds Ratio	95% Confidence Interval	*p*
Muscle atrophy (Presence)	1.8998	0.6596–5.4631	0.2347
NLR (High)	1.4942	0.6247–3.5743	0.3668

Abbreviation: NLR, neutrophil-to-lymphocyte ratio.

**Table 4 nutrients-12-01884-t004:** Logistic regression analysis with a stepwise variable selection for the factors associated with muscle atrophy.

Factors	Unit	Odds Ratio	95% Confidence Interval	*p*
NLR (High)	N/A	1.3341	1.1603–1.5721	0.0272
Age	1	1.2131	1.0587–1.3891	0.03608

Abbreviation: NLR, neutrophil-to-lymphocyte ratio.

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
