# Peer review of "Profiles Combining Muscle Atrophy and Neutrophil-to-Lymphocyte Ratio Are Associated with Prognosis of Patients with Stage IV Gastric Cancer"

_nutrients, 2020, doi:10.3390/nu12061884_

Round 1

Reviewer 1 Report

Thank you for the opportunity to review the manuscript “Profiles Combining Sarcopenia and Neutrophil-to-Lymphocyte Ratio Are Associated with Prognosis of Patients with Stage IV Gastric Cancer” by Shigeto K et al.

This is a very interesting single centre retrospective study analysing the impact of sarcopenia and high neutrophil-to-lymphocyte ratio in the prognosis of patients with stage IV gastric cancer. Combining these two markers is an innovative idea, and I found the manuscript to be clear and succinct. Overall, it is scientifically and technically good, I would suggest only minor improvements.

Concerns:

Page 3, Line 119: You evaluated visceral fat area and hepatic fat content, but there is no information in the Introduction/Discussion regarding these parameters, and their association with survival/prognosis. Are there any validated cut-off values for these parameters? Please consider adding information on that.

Page 4, Line 147: Please confirm if the following sentence is correct: “78.0% of subjects were female”. In the abstract and Table 1 it appears as 78% males.

Page 8, Line 181: Did you include other variables in the Cox regression model? For instance, age, performance status, location, albumin, HbA1c? Please explain why you didn’t.

Page 8, Line 193: Please explain the benefits of using a decision-tree algorithm in comparison with standard survival analysis.

Page 10, Line 239: “(…) patients in our study may be more severe (…)” Please change to “patients in our study may have a more advanced disease”.

Page 10, Line 250: Do you have any information on chemotherapy toxicity in these patients?

Page 10, Line 265: Keeping your results in mind what would be your future directions regarding this subject?

Table 1: In the section “Presence of distant metastasis (Yes/No)” there are two lines with different percentages, please clarify/correct.

Table 1: In the section “Presence of sarcopenia (Yes/No)” the results report 82.6% No, please correct this.

Tables 1 and 2: Tables 1 and 2 look very busy, maybe you could include the laboratory results such as haemoglobin, AST, (…) in a supplementary Table. In Table 2 consider keeping the results that were significant (LDH, albumin, HbA1c).

Author Response

To REVIEWRE 1

Thank you very much for your letter regarding our manuscript (nutrients-826629). We appreciate your comments, which have helped us to improve our manuscript. In line with your comments, please find below our point-by-point responses.

Comment 1) Page 3, Line 119: You evaluated visceral fat area and hepatic fat content, but there is no information in the Introduction/Discussion regarding these parameters, and their association with survival/prognosis. Are there any validated cut-off values for these parameters? Please consider adding information on that.

Answer: We appreciate for your comment. As the reviewer pointed out, we should have described information for visceral fat area and hepatic fat content throughout the manuscript. We have added following descriptions in each section of the revised manuscript:

1) In the introduction section, visceral adiposity and fatty liver are risk factors for poor prognosis of patients with various cancers [1, 2] (line 46-47).

2) In the materials and methods section, visceral adiposity was diagnosed as ≥ 100 cm2 of visceral fat area [3] (line 129-130). Fatty liver was diagnosed as liver-to-spleen ratios < 0.9 [4] (line 132-133).

3) In the results section, visceral adiposity and fatty liver were seen in 20% and 2% of patients in this study (line 164). There was no significant difference in the prevalence of visceral adiposity between the two groups. The prevalence of fatty liver was significantly higher in the Alive group compared to the Deceased group (line 181-183). Factors including visceral adiposity, and fatty liver were not selected by a stepwise variable selection in a Cox regression analysis (line 202-204). Factors including visceral adiposity, and fatty liver were not identified as a divergence variable in a decision-tree analysis (line 226-228).

4) In the Discussion section, visceral adiposity and fatty liver were not identified as an independent prognostic factor (line 270-271).

Comment 2) Page 4, Line 147: Please confirm if the following sentence is correct: “78.0% of subjects were female”. In the abstract and Table 1 it appears as 78% males.

Answer: We apologize for the typo. As you indicated, “78.0% of subjects were male”. We have corrected the typo in the revised manuscript (line 160).

Comment 3) Page 8, Line 181: Did you include other variables in the Cox regression model? For instance, age, performance status, location, albumin, HbA1c? Please explain why you didn’t.

Answer: We apologize for unclear description for the metrology and results of the Cox regression model. In this study, independent factors associated with the mortality were analyzed using a Cox regression analysis with a step-wise variable selection as previously described [5, 6]. Various factors were included in the analysis. However, following factors were not selected as variables for a Cox regression analysis by a stepwise variable selection: age, sex, BMI, location of gastric cancer, treatment regimen for gastric cancer, visceral adiposity, fatty liver, serum albumin level, and blood HbA1c level. Although NLR and sarcopenia were selected as variables for the Cox regression analysis by a stepwise variable selection, these factors were not statistically associated with mortality. We revised the metrology and results of the Cox regression model (line 153-154, 202-204).

Comment 4) Page 8, Line 193: Please explain the benefits of using a decision-tree algorithm in comparison with standard survival analysis.

Answer: We apologize for insufficient description for a decision-tree algorithm. A decision-tree algorithm is a data-mining technique that reveals a series of classification rules by identifying priorities. Thus, the benefit of using a decision-tree algorithm is that we can disclose a combination of important factors for mortality. Recently, a decision-tree algorithm has been used to identify the profiles associated with various patients with cancer [7-10]. We have added the explanation of a decision-tree algorithm in the revised manuscript (line 217-219).

Comment 5) Page 10, Line 239: “(…) patients in our study may be more severe (…)” Please change to “patients in our study may have a more advanced disease”.

Answer: We appreciate for your careful peer review. We have corrected the sentence to “patients in our study may have a more advanced disease” following your suggestion (line 279).

Comment 6) Page 10, Line 250: Do you have any information on chemotherapy toxicity in these patients?

Answer: We apologize that we did not provide information on chemotherapy toxicity. In this study, adverse event was assessed based on the National Cancer Institute Common Terminology Criteria for Adverse Events, version 4.0 [11]. Grade 3 of adverse event were seen in 74.3% of patients, and neutropenia was a major adverse event, which was seen in 35.8% of patients. There was no significant difference in the prevalence of grade 3 of adverse event and neutropenia between the Alive and Deceased groups. These data were added in the text and Tables of the revised manuscript (line 141-145, 162-163, 179-180, Tables 1 and 2). We deeply appreciate for your valuable comment, which have helped us to improve our manuscript.

Comment 7) Page 10, Line 265: Keeping your results in mind what would be your future directions regarding this subject?

Answer: We appreciate for your comment. As you pointed out, we did not describe the direction of future study. We revised the sentence as following: Thus, the prognostic importance of a profile combining sarcopenia and high NLR should be evaluated in a multicenter prospective cohort study should be conducted in a homogenous cohort of patients with stage IV gastric cancer (line 311-313).

Comment 8) Table 1: In the section “Presence of distant metastasis (Yes/No)” there are two lines with different percentages, please clarify/correct.

Answer: We appreciate for your careful peer review and apologize for the typo. We deleted the lower line and revised the description as following: 97.2%/2.8% (106/3) (Table 1).

Comment 9) Table 1: In the section “Presence of sarcopenia (Yes/No)” the results report 82.6% No, please correct this.

Answer: Again, we appreciate for your careful peer review and apologize for the typo. We revised the description as following: Presence of muscle atrophy (Yes/No); 82.6%/17.4% (90/19) (Table 1).

Comment 10) Tables 1 and 2: Tables 1 and 2 look very busy, maybe you could include the laboratory results such as haemoglobin, AST, (…) in a supplementary Table. In Table 2 consider keeping the results that were significant (LDH, albumin, HbA1c).

Answer: We appreciate for your comment. We agree with your comment and, in order to readability, all laboratory results in Table 1 were described in Supplementary Table 1. In Table 2, laboratory results with no significant difference between the groups were described in Supplementary Table 2 according to your suggestion (Table 1 and 2).

References

1    Xiao, J.; Mazurak, V.C.; Olobatuyi, T.A.; Caan, B.J., Prado, C.M. Visceral adiposity and cancer survival: a review of imaging studies. Eur J Cancer Care (Engl). 2018; 27: e12611.

2    Hwang, Y.C.; Ahn, H.Y.; Park, S.W., Park, C.Y. Nonalcoholic Fatty Liver Disease Associates With Increased Overall Mortality and Death From Cancer, Cardiovascular Disease, and Liver Disease in Women but Not Men. Clin Gastroenterol Hepatol. 2018; 16: 1131-7 e5.

3    Minamino, H.; Katsushima, M.; Yoshida, T.; Hashimoto, M.; Fujita, Y.; Shirakashi, M.; Yamamoto, W.; Murakami, K.; Murata, K.; Nishitani, K.; et al. Increased circulating adiponectin is an independent disease activity marker in patients with rheumatoid arthritis: A cross-sectional study using the KURAMA database. PLoS One. 2020; 15: e0229998.

4    Fujii, Y.; Nanashima, A.; Hiyoshi, M.; Imamura, N.; Yano, K., Hamada, T. Risk factors for development of nonalcoholic fatty liver disease after pancreatoduodenectomy. Ann Gastroenterol Surg. 2017; 1: 226-31.

5    Yamamura, S.; Kawaguchi, T.; Nakano, D.; Tomiyasu, Y.; Yoshinaga, S.; Doi, Y.; Takahashi, H.; Anzai, K.; Eguchi, Y.; Torimura, T.; et al. Profiles of advanced hepatic fibrosis evaluated by FIB-4 index and shear wave elastography in health checkup examinees. Hepatol Res. 2020; 50: 199-213.

6    Noda, Y.; Kawaguchi, T.; Korenaga, M.; Yoshio, S.; Komukai, S.; Nakano, M.; Niizeki, T.; Koga, H.; Kawaguchi, A.; Kanto, T.; et al. High serum interleukin-34 level is a predictor of poor prognosis in patients with non-viral hepatocellular carcinoma. Hepatol Res. 2019; 49: 1046-53.

7    Kawaguchi, T.; Tokushige, K.; Hyogo, H.; Aikata, H.; Nakajima, T.; Ono, M.; Kawanaka, M.; Sawada, K.; Imajo, K.; Honda, K.; et al. A Data Mining-based Prognostic Algorithm for NAFLD-related Hepatoma Patients: A Nationwide Study by the Japan Study Group of NAFLD. Sci Rep. 2018; 8: 10434.

8    Shimose, S.; Tanaka, M.; Iwamoto, H.; Niizeki, T.; Shirono, T.; Aino, H.; Noda, Y.; Kamachi, N.; Okamura, S.; Nakano, M.; et al. Prognostic impact of transcatheter arterial chemoembolization (TACE) combined with radiofrequency ablation in patients with unresectable hepatocellular carcinoma: Comparison with TACE alone using decision-tree analysis after propensity score matching. Hepatol Res. 2019; 49: 919-28.

9    Yang, C.C.; Su, Y.C.; Lin, Y.W.; Huang, C.I., Lee, C.C. Differential impact of age on survival in head and neck cancer according to classic Cox regression and decision tree analysis. Clin Otolaryngol. 2019; 44: 244-53.

10  Tanaka, T., Voigt, M.D. Decision tree analysis to stratify risk of de novo non-melanoma skin cancer following liver transplantation. J Cancer Res Clin Oncol. 2018; 144: 607-15.

11  Narayan, R.; Blonquist, T.M.; Emadi, A.; Hasserjian, R.P.; Burke, M.; Lescinskas, C.; Neuberg, D.S.; Brunner, A.M.; Hobbs, G.; Hock, H.; et al. A phase 1 study of the antibody-drug conjugate brentuximab vedotin with re-induction chemotherapy in patients with CD30-expressing relapsed/refractory acute myeloid leukemia. Cancer. 2020; 126: 1264-73.

Reviewer 2 Report

Sarcopenia is a very important topic in the panorama of malnutrition due to its association with poor oncological outcomes. Moreover, inflammation plays an important role in malnutrition and prognosis worsening.

About your paper, I have some comments.

Line 56: Reference 16 is old. In 2010 has been published a clinical definition and consensus diagnostic criteria for age-related sarcopenia by EWGSOP.

Line 20: Reference 20 refers to an elderly population. It can be add a reference including also younger adults.

Line 100: NLR, from literature data, is associated with an adverse OS in many solid tumours but also Glasgow Prognostic Score( GPS) is a powerful and well established parameter to rate systemic inflammation, the acute-phase proteins C-reactive protein and albumin (combined in GPS) are most clinically useful. Why didn't you consider using this score too?

Lines 114-118: According to EWGSOP definition, assessment of muscle strength and physical performance is required  to assess sarcopenia. The eduction of muscle mass (even if it has been used a direct gold standard technique ad imaging), alone this is not optimum, it add an information to malnutrition.

Line 121: The umbilical level corresponds to L3 (third lumbar vertebra, landmark used to evaluate body composition with CT)?

Line 147: 78% of subjects were MALE, not female, according with Table 1.

Author Response

To REVIEWRE 2

Thank you very much for your letter regarding our manuscript (nutrients-826629). We appreciate your comments, which have helped us to improve our manuscript. In line with your comments, please find below our point-by-point responses.

Comment 1) Line 56: Reference 16 is old. In 2010 has been published a clinical definition and consensus diagnostic criteria for age-related sarcopenia by EWGSOP.

Answer: As you pointed out, reference 16 is old and we have replace it to the up-dated definition of sarcopenia by the European Working Group on Sarcopenia in Older People 2 [1] following your suggestion (line 60).

Comment 2) Line 20: Reference 20 refers to an elderly population. It can be add a reference including also younger adults.

Answer: We appreciate for your comment. As you pointed out, reference 20 refers to an elderly population. In the revised manuscript, we have added a reference including also younger adults following your suggestion [2] (line 62-64).

Comment 3) Line 100: NLR, from literature data, is associated with an adverse OS in many solid tumours but also Glasgow Prognostic Score (GPS) is a powerful and well established parameter to rate systemic inflammation, the acute-phase proteins C-reactive protein and albumin (combined in GPS) are most clinically useful. Why didn't you consider using this score too?

Answer: We appreciate for your valuable comment. As you pointed out, GPS is a well-established parameter and is a useful scoring system to evaluate systemic inflammation in patients with cancer [3]. However, GPS is consisted of 3 ordinal variables (Point 0, 1, and 3) [3]. In general, ordinal variable does not have fixed unit of measurement and gaps between the variables are not equal, indicating the difficulty for various clinical situation [4]. While NLR is a continuous variable, which is able to reflect to the wild rage of clinical situation [4]. In fact, NLR has been reported to be superior to GPS to predict the recurrence after curative surgery for patients with Stage II/III gastric cancer [5]. In the revised manuscript, we have added the description of GPS (line 47-48, line 53-55). We deeply appreciate for your valuable comment, which have helped us to improve our manuscript.

Comment 4) Lines 114-118: According to EWGSOP definition, assessment of muscle strength and physical performance is required to assess sarcopenia. The eduction of muscle mass (even if it has been used a direct gold standard technique ad imaging), alone this is not optimum, it add an information to malnutrition.

Answer: We totally agree with your comment. Since we did not assess muscle function, we should not use “sarcopenia” in this study. In the revised manuscript, we have replaced “sarcopenia” to “muscle atrophy” throughout the manuscript.

Comment 5) Line 121: The umbilical level corresponds to L3 (third lumbar vertebra, landmark used to evaluate body composition with CT)?

Answer: We appreciate for your comment. As you pointed out, we evaluated skeletal muscle mass using CT imaging of the L3 level. While, CT imaging of the umbilical level was used to evaluate visceral fat area as previously described [6]. The umbilical level is not identical to L3 level and corresponds to approximately the level of L4 and L5; however, visceral fat area at the umbilical level is known to significantly correlates with total visceral abdominal fat volume [7-9] and is generally used to evaluate visceral fat area [10, 11]. We added the explanation in the revised manuscript (line 127-128).

Comment 6) Line 147: 78% of subjects were MALE, not female, according with Table 1.

Answer: We apologize for the typo. As you indicated, “78.0% of subjects were male”. We have corrected the typo in the revised manuscript (line 160).

References

1    Cruz-Jentoft, A.J.; Bahat, G.; Bauer, J.; Boirie, Y.; Bruyere, O.; Cederholm, T.; Cooper, C.; Landi, F.; Rolland, Y.; Sayer, A.A.; et al. Sarcopenia: revised European consensus on definition and diagnosis. Age Ageing. 2019; 48: 16-31.

2    Zhuang, C.L.; Huang, D.D.; Pang, W.Y.; Zhou, C.J.; Wang, S.L.; Lou, N.; Ma, L.L.; Yu, Z., Shen, X. Sarcopenia is an Independent Predictor of Severe Postoperative Complications and Long-Term Survival After Radical Gastrectomy for Gastric Cancer: Analysis from a Large-Scale Cohort. Medicine (Baltimore). 2016; 95: e3164.

3    McMillan, D.C. The systemic inflammation-based Glasgow Prognostic Score: a decade of experience in patients with cancer. Cancer Treat Rev. 2013; 39: 534-40.

4    Cash, E., Boktor, S.W. Understanding Biostatistics Interpretation.  StatPearls. Treasure Island (FL) 2020.

5    Tanaka, H.; Tamura, T.; Toyokawa, T.; Muguruma, K.; Miki, Y.; Kubo, N.; Sakurai, K.; Hirakawa, K., Ohira, M. Clinical Relevance of Postoperative Neutrophil-Lymphocyte Ratio (NLR) to Recurrence After Adjuvant Chemotherapy of S-1 for Gastric Cancer. Anticancer Res. 2018; 38: 3745-51.

6    Koya, S.; Kawaguchi, T.; Hashida, R.; Hirota, K.; Bekki, M.; Goto, E.; Yamada, M.; Sugimoto, M.; Hayashi, S.; Goshima, N.; et al. Effects of in-hospital exercise on sarcopenia in hepatoma patients who underwent transcatheter arterial chemoembolization. J Gastroenterol Hepatol. 2019; 34: 580-8.

7    Tokunaga, K.; Matsuzawa, Y.; Ishikawa, K., Tarui, S. A novel technique for the determination of body fat by computed tomography. Int J Obes. 1983; 7: 437-45.

8    Sjostrom, L.; Kvist, H.; Cederblad, A., Tylen, U. Determination of total adipose tissue and body fat in women by computed tomography, 40K, and tritium. Am J Physiol. 1986; 250: E736-45.

9    Kvist, H.; Chowdhury, B.; Sjostrom, L.; Tylen, U., Cederblad, A. Adipose tissue volume determination in males by computed tomography and 40K. Int J Obes. 1988; 12: 249-66.

10  Demura, S., Sato, S. Nonlinear relationships between visceral fat area and percent regional fat mass in the trunk and the lower limbs in Japanese adults. Eur J Clin Nutr. 2008; 62: 1395-404.

11  Ho, L.C.; Yen, C.J.; Chao, C.T.; Chiang, C.K.; Huang, J.W., Hung, K.Y. Visceral fat area is associated with HbA1c but not dialysate-related glucose load in nondiabetic PD patients. Sci Rep. 2015; 5: 12811.

Reviewer 3 Report

Dear Author, 

This is an interesting study which aimed to assess combined prognostic of sarcopenia and NLR on overall survival in stage IV gastric cancer 

However, I have several comments about your manuscript: 

1/ Abstract

female/male 24/85: please change for % as follows "female/male 22%/78%"

2/ INTRODUCTION: 

  • line 56: reference 16 is very old...To date, numerous consensuses exist. I suggest the last European consensus (2019) or maybe the Asian consensus which is close to your population study. Moreover, you state that sarcopenia in various cancers is frequent. This is true. Could you provide a reference for that with a prevalence ? I suggest the review by Pamoukdjian et al. Clin Nutr. 2018 Aug;37(4):1101-1113. 

3/ METHODS:

  • could you provide more for the study design ? Is this study single or multi center ?
  • Sarcopenia is the associated loss of skeletal muscle and function. Here, you measure only muscle mass. Could you state that it is not sarcopenia stricto sensu ? It is better to write "Patients were classified as function of low skeletal muscle mass or not according to the SMI" or something similar. Please correct this thorough the manuscript beginning by title (i.e. low skeletal muscle mass instead of sarcopenia) 
  • statistical analysis: you performed a combined analysis of low skeletal muscle AND NLR. It is important for readers to explain why you did it? for example, did you assess interaction terms between muscle mass X NLR ? Moreover, did you assess model assumptions of Cox regression (i.e. proportional risk). 

4/ RESULTS: 

  • it seems to have a typo line 147: you wrote 78% female but Table 1 is 22%. Please clarify this. 

5/ DISCUSSION: 

I think there is a probable interaction between muscle mass and NLR. It is important to assess this. If you show a such interaction, it would be very interesting to discuss this because NLR is an indirect marker of poor nutrition status which we know it is a cause of "sarcopenia" 

Best regards 

Author Response

To REVIEWRE 3

Thank you very much for your letter regarding our manuscript (nutrients-826629). We appreciate your comments, which have helped us to improve our manuscript. In line with your comments, please find below our point-by-point responses.

Comment 1) 1/ Abstract, female/male 24/85: please change for % as follows "female/male 22%/78%".

Answer: We appreciate for your careful peer review and apologize for the typo. We have revised the description as "female/male 22%/78%" following your suggestion (line 22).

Comment 2) 2/ INTRODUCTION: line 56: reference 16 is very old...To date, numerous consensuses exist. I suggest the last European consensus (2019) or maybe the Asian consensus which is close to your population study. Moreover, you state that sarcopenia in various cancers is frequent. This is true. Could you provide a reference for that with a prevalence? I suggest the review by Pamoukdjian et al. Clin Nutr. 2018 Aug;37(4):1101-1113.

Answer: As you pointed out, reference 16 is old and we have replace it to the up-dated definition of sarcopenia by the European Working Group on Sarcopenia in Older People 2 and Asian Working Group for Sarcopenia [1, 2] following your suggestion (line 60). In addition, we have described the prevalence of sarcopenia by referring the review article by Pamoukdjian et al. [3]; Sarcopenia is seen in approximately 40% of patients with various types of advanced cancers [3] (line 60-61).

Comment 3) 3/ METHODS:

Comment 3-1) could you provide more for the study design? Is this study single or multi center?

Answer: We apologize for unclear description for our study design. In the revised manuscript, we have described as following: This is a single center retrospective observational study to investigate prognostic profiles for patients with stage IV gastric cancer (line 74-75).

Comment 3-2) Sarcopenia is the associated loss of skeletal muscle and function. Here, you measure only muscle mass. Could you state that it is not sarcopenia stricto sensu? It is better to write "Patients were classified as function of low skeletal muscle mass or not according to the SMI" or something similar. Please correct this thorough the manuscript beginning by title (i.e. low skeletal muscle mass instead of sarcopenia).

Answer: We totally agree with your comment. Since we did not assess muscle function, we should not use “sarcopenia” in this study. In the revised manuscript, we have replaced “sarcopenia” to “muscle atrophy” throughout the manuscript.

Comment 3-3) statistical analysis: you performed a combined analysis of low skeletal muscle AND NLR. It is important for readers to explain why you did it? for example, did you assess interaction terms between muscle mass X NLR? Moreover, did you assess model assumptions of Cox regression (i.e. proportional risk). 

Answer: We appreciate for your comment. In this study, none of single factor was associated with mortality in the Cox regression analysis. Therefore, we performed a decision-tree analysis to find out combinations of factors associated with mortality. Thus, the combination of muscle atrophy and high NLR is based on the results of the decision-tree algorithm for mortality (Figure 1). The analysis revealed that a combination of muscle atrophy and high NLR is the worst profile associated with the mortality. We have added the explanation in the revised manuscript (line 282-284).

            In the revised manuscript, we further investigated the impact of profiles according to the results of decision-tree algorithm for survival. Profile 2 and 3 were identified as independent factor for mortality in the Cox regression analysis (Profile 2, OR 2.9321,95%CI 1.0088-8.5220, P=0.9481; Profile 3, OR 6.8024, 95%CI 1.8069-25.6093, P=0.0046). We have also added these results in the revised manuscript. Again, we deeply appreciate for your valuable comment, which have helped us to improve our manuscript (line 236-239).

Comment 4) 4/ RESULTS: it seems to have a typo line 147: you wrote 78% female but Table 1 is 22%. Please clarify this. 

Answer: We apologize for the typo. As you indicated, “78.0% of subjects were male”. We have corrected the typo in the revised manuscript (line 160).

Comment 5) DISCUSSION: I think there is a probable interaction between muscle mass and NLR. It is important to assess this. If you show a such interaction, it would be very interesting to discuss this because NLR is an indirect marker of poor nutrition status which we know it is a cause of "sarcopenia" 

Answer: We totally agree with your comment. Following your suggestion, we have performed a Cox regression analysis and found that independent factors associated with muscle atrophy were high NLR (OR 1.3341 [95%CI:1.1603-1.5721], P=0.0272) and age (Unit 1, OR 1.2131 [95%CI:1.0587-1.3891], P=0.03608). As the reviewer pointed out, NLR is an indirect marker of poor nutrition status, which leads sarcopenia [4, 5]. In addition, Borges et al. recently reported that high NLR values were independently associated with sarcopenia in hospitalized patients with cancer [6]. Taken together, these data indicated a probable interaction between muscle mass and NLR. We have added these results in the revised manuscript (line 252-254, line 300-305, Table 4). We deeply appreciate for your valuable comment, which have helped us to improve our manuscript.

References

1  Cruz-Jentoft, A.J.; Bahat, G.; Bauer, J.; Boirie, Y.; Bruyere, O.; Cederholm, T.; Cooper, C.; Landi, F.; Rolland, Y.; Sayer, A.A.; et al. Sarcopenia: revised European consensus on definition and diagnosis. Age Ageing. 2019; 48: 16-31.

2  Chen, L.K.; Woo, J.; Assantachai, P.; Auyeung, T.W.; Chou, M.Y.; Iijima, K.; Jang, H.C.; Kang, L.; Kim, M.; Kim, S.; et al. Asian Working Group for Sarcopenia: 2019 Consensus Update on Sarcopenia Diagnosis and Treatment. J Am Med Dir Assoc. 2020; 21: 300-7 e2.

3  Pamoukdjian, F.; Bouillet, T.; Levy, V.; Soussan, M.; Zelek, L., Paillaud, E. Prevalence and predictive value of pre-therapeutic sarcopenia in cancer patients: A systematic review. Clin Nutr. 2018; 37: 1101-13.

4  Sato, Y.; Gonda, K.; Harada, M.; Tanisaka, Y.; Arai, S.; Mashimo, Y.; Iwano, H.; Sato, H.; Ryozawa, S.; Takahashi, T.; et al. Increased neutrophil-to-lymphocyte ratio is a novel marker for nutrition, inflammation and chemotherapy outcome in patients with locally advanced and metastatic esophageal squamous cell carcinoma. Biomed Rep. 2017; 7: 79-84.

5  Diaz-Martinez, J.; Campa, A.; Delgado-Enciso, I.; Hain, D.; George, F.; Huffman, F., Baum, M. The relationship of blood neutrophil-to-lymphocyte ratio with nutrition markers and health outcomes in hemodialysis patients. Int Urol Nephrol. 2019; 51: 1239-47.

6  Borges, T.C.; Gomes, T.L.; Pichard, C.; Laviano, A., Pimentel, G.D. High neutrophil to lymphocytes ratio is associated with sarcopenia risk in hospitalized cancer patients. Clin Nutr. 2020.

Round 2

Reviewer 3 Report

Dear Author,

Thanks for your revised version

I have no further comments

Best regards